# Strategies to Make Telemedicine a Friend, Not a Foe, in the Provision of Accessible and Equitable Cancer Care

**DOI:** 10.3390/cancers15215121

**Published:** 2023-10-24

**Authors:** Brook A. Calton, Sarah Nouri, Carine Davila, Ashwin Kotwal, Carly Zapata, Kara E. Bischoff

**Affiliations:** 1Department of Medicine, Division of Palliative Medicine and Geriatrics, The Massachusetts General Hospital, Boston, MA 02114, USA; cdavila@mgh.harvard.edu; 2Department of Medicine, Division of Palliative Medicine, University of California, San Francisco, CA 94720, USA; sarah.nouri@ucsf.edu (S.N.); carly.zapata@ucsf.edu (C.Z.); kara.bischoff@ucsf.edu (K.E.B.); 3Department of Medicine, Division of Geriatrics, University of California, San Francisco, CA 94720, USA; ashwin.kotwal@ucsf.edu; 4Geriatrics, Palliative, and Extended Care Service Line, San Francisco Veterans Affairs Medical Center, San Francisco, CA 94121, USA

**Keywords:** telemedicine, telehealth, access, equity, disparities, barriers

## Abstract

**Simple Summary:**

In this Perspective, we explore the benefits of telemedicine in extending cancer care to patients who face difficulties due to functional limitations, high symptom burden, and financial or geographic constraints. We also acknowledge the risk that telemedicine may worsen existing healthcare inequalities for patients who already face systemic disadvantages. To address these concerns and increase the likelihood that telemedicine is a tool to improve equity, we suggest practical strategies and policy recommendations aimed at ensuring high-quality, accessible telemedicine-based cancer care.

**Abstract:**

Telemedicine has the potential to improve access to cancer care, particularly for patients with functional limitations, high symptom burdens, or financial or geographic constraints. However, there is also a risk that telemedicine can widen healthcare disparities among patients facing systemic disadvantages like those with technological barriers, poor digital literacy, older age, or non-English language preferences. To optimize telemedicine usage, we must implement practical strategies like video onboarding programs, user-friendly technology platforms, optimizing the clinician’s environment, and best practices for using interpreters. Policy changes such as state licensing requirements, controlled substance prescribing requirements, and payment parity are also crucial. This Perspective highlights these practical strategies and policy recommendations to ensure accessible and equitable cancer care augmented by telemedicine.

## 1. Introduction

During the COVID-19 pandemic, the use of video telemedicine visits skyrocketed, enabling patients to access essential healthcare, including cancer care, while social distancing [1]. Post pandemic, telemedicine will remain an essential modality to deliver patient-centric, flexible cancer care [1]. Telemedicine has the potential to increase access to oncologic consultations, second opinions, follow-up care, and supportive care services (palliative care, survivorship, psycho-oncology, nutrition, etc.), particularly for functionally limited, highly symptomatic individuals or those with limited access to cancer care in their local communities. On the other hand, an evolving body of literature has highlighted disparities in the access to and usability of telemedicine among populations experiencing systemic disadvantages, including older adults, people of color, and those with a preferred language other than English [2,3]. Given the possibility that video telemedicine can improve access to cancer care and the risk that it can exacerbate disparities, in this Perspective, we outline practical strategies and policy recommendations to optimize the use of telemedicine so that it can be a friend, not a foe, in improving access and equity in cancer care.

## 2. How Telemedicine Can Exacerbate Healthcare Disparities

It is estimated that as many as one in four US adults may not have the technology access or digital literacy skills needed to engage in telemedicine [4]. Regarding technology access, while the number of people with smartphones is increasing, only 43% of people 75 and older have smartphones [5]. Moreover, broadband access, which facilitates better video quality than cellular data, is as low as 57% among low-income households and significantly lower among Black and Hispanic/Latino adults compared to White adults [4,5]. In a recent study of 6006 patients newly diagnosed with cancer early in the COVID-19 pandemic, patients in the highest socioeconomic status index quartile were 31% more likely to use telemedicine within 30 days of cancer diagnosis compared with patients in the lowest socioeconomic status index quartile [6].

In addition, older adults are more likely to have limited digital literacy, or the ability to find, evaluate, and communicate information using the internet or other digital platforms. And even among people with access to telemedicine, usability is low for people with certain disabilities, such as hearing or visual impairment [4,7,8].

## 3. How Telemedicine Can Expand Access to Care and Reduce Disparities

Telemedicine has been shown to extend care to some populations that would not otherwise have access, including patients with cancer and substantial debility, who live a long distance from their oncology care, or who experience financial burden due to the travel expenses and time away from work required to be seen in-person. A review article published in 2021 found telemedicine improved access to care, rendered improved health outcomes in several areas of medicine beyond cancer (e.g., stroke and heart failure management), and reduced geographical barriers to care [9]. In a recent article of nearly 12,000 patients with cancer, patient out-of-pocket costs were on average USD 147 to 186 cheaper for telemedicine compared to in-person visits [10]. A study performed in an outpatient palliative medicine practice at an academic medical center that serves a patient population of predominantly advanced cancer patients from across Northern California found that there were far fewer disparities by race, ethnicity, preferred language, and insurance status in who was able to access care for people seen by telemedicine compared to those who were seen in person [11]. The authors hypothesized that this was because telemedicine obviates the need for seriously ill patients and their caregivers to undertake costly, time-consuming, and burdensome travel to the medical center. Telemedicine also makes it easier for caregivers who have other work or family commitments or live far away to participate in medical visits.

Given both the risk that video telemedicine could exacerbate disparities and that, particularly for seriously ill and debilitated individuals, telemedicine can improve access to care, here, we outline practical strategies and policy recommendations to optimize the use of telemedicine in the care of patients with cancer.

## 4. Practical Strategies for Digital Navigation

### 4.1. Video Onboarding

A thoughtful onboarding program is critical to reduce disparities when using telemedicine. Patient educational resources that walk people through the steps to prepare for and launch a telemedicine visit should be available as printed handouts and electronically in multiple languages. Many health systems have invested in creating videos showing patients how to onboard. Some patients will be able to onboard with these educational materials alone, while others will require one-on-one support. Trained administrative staff (who are either part of a clinic team or part of a telemedicine resource center funded by a health system) should be available to onboard patients who need help by phone or in person. These sessions can also help identify patient-specific needs, like sensory impairment or the need for an interpreter, that can be addressed [12]. If a patient lacks the ability or technology to launch a video visit, caregivers, family, friends, neighbors, and home-based resources (e.g., the home health team) should be engaged to see if they can provide the necessary technology (e.g., smartphone) and/or help with set-up in advance of a telemedicine visit.

In addition, a process should be in place to denote a patient’s telemedicine readiness in the medical record. Checklists are available that assess patients’ comfort and access to the technology required to conduct a video telemedicine visit [13,14]. The accurate assessment and documentation of telemedicine readiness allows health systems and clinics to identify and troubleshoot barriers to their patient population’s ability to access telemedicine-based care.

### 4.2. Technology Considerations

Using a secure, HIPAA-compliant platform that minimizes the steps needed to log onto a telemedicine visit is key, particularly for new users or those unfamiliar with digital technologies. Some platforms make use of a link or URL that can be sent through a cellular phone text message or e-mail. These links directly connect patients to the visit without the need to download applications or log into a patient’s online medical record portal, which can be barriers for some patients. This “one-step login” allows patients to navigate to the visit effectively regardless of technical proficiency and saves time during clinical encounters.

Telemedicine platforms that offer closed-captioning (written transcription of the conversation in real-time) or a chat function can be helpful for patients who are hard of hearing or unable to speak (e.g., a patient with head and neck cancer with a laryngectomy). For people who are hard of hearing, amplified headphones can help (in the case of using headphones, family members joining the visit may need to log on from additional devices to participate). For people with limited access to digital technologies, the Federal Lifeline program is a resource for eligible low-income patients to receive subsidized phone and broadband services [15].

To prepare for the technological difficulties that will inevitably arise at some frequency before or during telemedicine visits, patients should be given an office phone number to call (answered live) where they can receive help troubleshooting their technology. Training clinicians on how to coach their patients through common technology challenges (e.g., difficulty connecting to audio in Zoom) is also worthwhile to maximize success.

### 4.3. Optimize the Telemedicine Visit Environment

Clinicians are responsible for ensuring they are in a private space that is quiet and well lit before the telemedicine visit. Clinicians are encouraged to “be the director” and confirm both the clinician and patient can hear and see each other well throughout the visit. Simple things like asking a patient to relocate to a quieter or better-lit space can dramatically improve the quality of the visit and communication and minimize frustration by both parties. Optimizing the telemedicine visit environment on both the clinician and patient end is particularly key for patients who are hard of hearing or have difficulty with attention.

Clinicians should ensure they know where the patient is located during the visit and ask if there are others present in that space. This is important to ensure confidentiality and maximize comfort given the sensitive nature of many conversations in cancer care. Assuming the patient is agreeable, a clinician can also request that a caregiver, who may be a valuable source of information and/or benefit from support, move to sit closer to the patient so both can be viewed on the video screen and participate in the visit. Practices and health systems should consider adopting and/or developing standardized training programs on telemedicine best practices to support high-quality telemedicine encounters.

### 4.4. Use of Interpreters

In the United States, more than 20% of individuals speak a language other than English at home, and nearly 1 in 12 speak English less than fluently [4,16,17]. It is crucial that individuals with limited English proficiency have access to reliable and timely interpreters to ensure equitable access to video-based healthcare. To achieve this, a clear process for including interpreters in telemedicine visits must be established. The first step is to identify the patient’s preferred spoken language and accurately reflect this information in the medical record. It is recommended to always use a professional medical interpreter rather than relying on bilingual family members. Depending on the clinical practice, video or phone interpreters may be used. Administrative staff can arrange for interpreters to be present during visits, and clinicians should be able to easily add interpreters during ongoing visits.

During visits, it is important to have the interpreter introduce themselves and explain their role to the patient. Clinicians should speak in shorter sentences or phrases and allow pauses for interpretation to ensure accuracy. Utilizing teach-back can help assess understanding, and debriefing with the interpreter after the visit can provide additional insight. It is helpful to request a continuity in interpreters for a patient’s care and to document their name and affiliation in the medical notes. To accommodate longer visits with interpreters, some clinical practices reserve additional time and use prolonged-visit billing codes. This is particularly important for initial visits to establish rapport and subsequent visits involving serious illness conversations and complex decision making.

## 5. Policy Recommendations

### 5.1. State Licensing

An essential element of expanding telemedicine access for patients with cancer is the consideration of the impact of licensing rules and the potential barriers they may pose for patients and providers. Currently, aside from the Veterans Affairs Health Care System, prescribing physicians need to be licensed in the state where the patient is physically located during the video visit (this rule was paused during the pandemic) [18]. This is straightforward for in-person visits; however, telemedicine enables a physician to see patients who are in many states. This is particularly true for cancer centers with a large catchment area in smaller states. Requiring clinicians to hold a state license in the state where their patient is located during the visit limits access to telemedicine cancer care and is a logistical and financial burden on clinicians and health systems. Ongoing initiatives aimed at achieving cross-state licensing (e.g., The Interstate Medical Licensure Compact for Physicians) are a positive step forward [19]. Medical boards could also develop waivers for state licensure for specific specialties that are not readily available in certain states, to improve equitable access to specialists via telemedicine.

### 5.2. Prescribing Requirements

Before the COVID-19 pandemic, the DEA restricted prescriptions of controlled substances to those patients who have had a face-to-face visit with the prescribing provider [20]. On 10 October 2023, the DEA announced an extension of telemedicine flexibilities that allow clinicians to continue to prescribe controlled substances without an in-person face-to-face visit (telemedicine sufficient) through 31 December 2024 [21]. The reinstatement of face-to-face restrictions is under active consideration by the DEA.

Resuming face-to-face requirements for controlled-substance prescribing would place a significant burden on patients with advanced and/or symptomatic cancer who may experience difficulty getting to a clinic for in-person visits and rely on controlled substances, including opioids for cancer-related pain or benzodiazepines for anxiety and nausea, to manage symptoms and preserve their quality and life [22,23]. Video telemedicine visits can make it easier for patients to be seen by a provider when they have severe symptoms, but if they are not able to obtain medications to address those symptoms, the utility of that improved access will be limited. In light of these concerns, we advocate that, at a minimum, certain populations of patients with cancer, including those receiving palliative or hospice care, should be able to receive controlled substances through telemedicine-based care without the requirement of an in-person visit.

### 5.3. Payment Parity

To ensure fair access to telemedicine, it is important to have payment parity, where the reimbursement rates for telemedicine are equal to those for in-person visits. This is especially crucial for small practices and underserved communities who may find it difficult to offer telemedicine if the reimbursement rates are much lower [24]. During the COVID-19 pandemic, the Center for Medicare and Medicaid Services (CMS) granted payment parity, which has been extended until the end of 2024. However, payment parity laws vary by state outside of the CMS. Some states, such as California and Massachusetts, have permanent telehealth payment parity laws that mandate equivalent reimbursement for telemedicine visits from commercial insurers. In other states, reimbursement rates are determined by payor contracts. Ongoing research into the quality of care provided by telemedicine and its actual costs compared to in-person visits will continue to inform this complex issue.

## 6. Conclusions

The enduring presence of telemedicine in healthcare and cancer care is undeniable. While we embrace its benefits including the potential to increase access for some patients, it is also important to address and strive to minimize the existing disparities in its utilization. This requires a comprehensive approach including actions at the level of the patient, clinician, practice, health system, and state and federal policy. Readers are encouraged to choose at least one strategy highlighted here to begin implementing today, while also advocating for longer-term system and policy solutions to maximize equitable access to cancer care by telemedicine.

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
