# Peer review of "Strategies to Make Telemedicine a Friend, Not a Foe, in the Provision of Accessible and Equitable Cancer Care"

_cancers, 2023, doi:10.3390/cancers15215121_

Round 1

Reviewer 1 Report

This perspective piece offers some ideas for ensuring equitable and functional telehealth for cancer care. It is well written, making it an "easy read" and the authors make cogent suggestions with references provided for each action.

I have only two minor edits to suggest:

Line 73 appears to be missing a word: "...rendered improved health outcomes in several areas of medicine beyond (e.g., stroke..." should likely read, "...medicine beyond cancer (e.g. stroke..."

Line 191 -192 "including opioids for cancer-related pain or benzodiazepines for nausea."  In my experience benzodiazpaines are used to treat nausea and I wonder if the authors meant to say dyspnea or perhaps anxiety.

Author Response

Thank you so much for reviewing the article and for your edits. Both of your edits have been made.  Regarding benzodiazepines, they can be used for anticipatory nausea and also for anxiety so we added anxiety into the text.  Thanks for this suggestion.

Reviewer 2 Report

Calton et al underscore various important facets to be taken into account when applying telemedicine. Nonetheless, it is essential to acknowledge the significance of factors like cybersecurity, the delivery and consistency of care, the education and skill development of healthcare practitioners, and the evaluation of telemedicine solutions.

The article, on the whole, is comprehensible.

Author Response

Thank you for your comment.  We have added several sentences into the manuscript to capture your important points:

Lines 148-150:"Practices and health systems should consider adopting and/or developing standardized training programs for clinicians on telemedicine best practices to support high-quality telemedicine encounters". 

Line 113: "Using a secure-HIPAA compliant platform"...

Reviewer 3 Report

An excellent contribution! One question: on p. 1,, line 38: is it social distancing? or socially distancing? (I am not a native English speaking person).

Author Response

Thank you for reviewing the article.  I believe it is "social distancing" and therefore have left the article as is.  

https://www.hopkinsmedicine.org/health/conditions-and-diseases/coronavirus/coronavirus-social-distancing-and-self-quarantine

Round 2

Reviewer 2 Report

Thank you for your reply